# Metabolic versatility and nitrate reduction pathways of a new thermophilic bacterium of the *Deferrivibrionaceae*: *Deferrivibrio metallireducens* sp. nov isolated from hot sediments of Vulcano Island, Italy

**Grégoire Galès** [ID]*, **Mélanie Hennart** [ID], **Maverick Hannoun, Anne Postec, Gaël Erauso***

Aix-Marseille Université, Université du Sud Toulon-Var, CNRS/INSU, IRD, MIO, UM 110, Marseille, France

* gregoire.gales@univ-amu.fr (GG); gael.erauso@univ-amu.fr (GE)

## Abstract

A novel thermophilic (optimum growth temperature ~60 °C) anaerobic Gram-negative bacterium, designated strain V6Fe1$^T$, was isolated from sediments heated by the hydrothermal circulation of the Aeolian Islands (Vulcano, Italy) on the seafloor. Strain V6Fe1$^T$ belongs to the recently described family *Deferrivibrionaceae* in the phylum *Deferribacterota*. It grows chemoorganotrophically by fermentation of proteinaceous substrates and organic acids or by respiration of organic compounds using fumarate, nitrate, Fe(III), S°, and Mn(IV) as electron acceptors. The strain V6Fe1$^T$ can also grow chemolithoautotrophically using $H_2$ as an electron donor and nitrate, nitrous oxide, Fe(III), Mn(IV), or sulfur as an electron acceptor. Stable isotope probing showed that V6Fe1$^T$ performs denitrification with nitrate reduction to dinitrogen and Dissimilatory Nitrate Reduction to Ammonium (DNRA). Culture experiments with RT-qPCR analysis of target genes revealed that strain V6Fe1$^T$ performs DNRA with the nitrite reductase formate-dependent NrfA and denitrification with an Hcp protein and other redox partners yet to be identified. Genomic analysis and experimental data suggest that strain V6Fe1$^T$ performs autotrophic carbon fixation *via* the recently discovered reversed oxidative TCA cycle (roTCA cycle). Based on genomic (ANI) and phenotypic properties, strain V6Fe1$^T$ (= DSM 27501$^T$ = JCM 39088$^T$) is proposed to be the type strain of a novel species named *Deferrivibrio metallireducens*.

## 1. Introduction

Vulcano is the southernmost of the seven islands forming the Aeolian archipelago. This island harbors a developed shallow hydrothermal system. It is where the first hyperthermophilic marine archaeon, *Pyrodictium occultum,* was isolated [1], leading to extensive studies of the geochemical and microbiological properties of this environment [2–4]. Several thermophilic and hyperthermophilic microorganisms have been isolated from Vulcano, including members of the *Bacteria* like *Aquifex aeolicus* [5] and *Thermotoga maritima* [6], members of the *Archaea* like *Archaeoglobus fulgidus* [7], *Ferroglobus placidus* [8], *Palaeococcus hegelsonii* [9],

**Data availability statement:** The GenBank/EMBL/DDBJ accession number for the complete genome sequence of strain V6Fe1T is CP063375.

**Funding:** The project X-TREMOPHILES (coordinator G. Erauso, 2017-2019) was founded by the program X-Life of the Interdisciplinarity Mission (MITI) of the CNRS (France).

**Competing interests:** The authors have declared that no competing interests exist.

*Thermococcus celer* [10], *Pyrococcus furiosus* [11], *Pyrodictium occultum* [1], and *Staphylothermus marinus* [11].

Nitrate reducers are commonly found within hydrothermal systems [12–14]. Thermodynamical calculation shows that nitrate may be one of the most suitable terminal electron acceptors for anaerobic respiration by chemolithotrophic or chemoorganotrophic lifestyles in Vulcano hydrothermal systems [12]. The energy yield of nitrate reduction was shown to be comparable to that of oxidation reaction with $O_2$. Two microbial metabolic pathways for nitrate reduction are known: denitrification, meaning that nitrate is reduced to $N_2$ or some intermediates like NO or $N_2O$, or Dissimilatory Nitrate Reduction to Ammonium (DNRA) in which nitrate is reduced to ammonium. Most organisms rely on reactive forms of nitrogen for growth, such as ammonium and nitrate, as they cannot fix atmospheric or dissolved $N_2$. Reactive forms of nitrogen are often growth-limiting nutrients in environmental conditions [15]. The fate of nitrate after being reduced is then critical, as ammonium produced during DNRA is still available in the environment, whereas $N_2$ is not. These processes are also of climate concern, as denitrification substantially releases the greenhouse intermediate $N_2O$ [16]. Nitrate reducers isolated from oceanic hydrothermal systems usually reduce nitrate to ammonium using sulfide or Fe(II) as an electron donor, like *Thermosulfuriphilus ammonigenes* [17], *Dissulfuribacter thermophilus* [18], most of the *Zetaproteobacteria* [19], *Caldithrix abyssi* [20], *Nautilia nitratireducens* [21]. Others reduce nitrate to nitrite, like *Vulcanibacillus modesticaldus* [22]. A few nitrate reducers isolated from these environments perform complete denitrification, like *Sulfurimonas paralvinellae* [23], *Nitratiruptor tergarcus*, and *Nitratifractor salsuginis* [24], which belong to the *Epsilonproteobacteria*. Nitrate reduction metabolism has been investigated in Filamentous Large Sulfide-Oxidizing Bacteria (FLSB) of deep hydrothermal systems like the Guaymas basin [13]. They were shown to be able to reduce their intracellular nitrate stocks to nitrogen gas and ammonium, depending on the concentration of electron donor (sulfide), by using stable isotope probing experiments and genome analysis [13].

Here, we describe a novel thermophilic strain V6Fe1T, isolated from the Vulcano hydrothermal system, belonging to the *Deferribacterota* phylum and the recently described *Deferrivibrionaceae* family, closely related to the *Deferribacteraceae* family. Cultivated representatives of the *Deferribacteraceae* and *Deferrivibrionaceae* families are thermophilic anaerobes. Most *Deferribacteraceae* use electron acceptors such as ferric iron and nitrate. Fermentative growth within *Deferribacteraceae* is rare and restricted to *D. desulfuricans* [25] and *P. organivorans* [26]. Autotrophic growth occurs in *D. autotrophicus* [27] and *D. abyssi* [28]. Strain V6Fe1T shows a unique metabolic versatility within *Deferribacteraceae* since it ferments various organic substrates, performs anaerobic respiration using various electron acceptors (ferric iron, manganese, elemental sulfur, nitrate), and grows lithoautotrophically with $H_2$ as an electron donor. Contrary to its closest relatives, strain V6Fe1T performs denitrification and DNRA, depending on the nature of the electron donor.

## 2.  Materials and methods

### 2.1  Geological setting and sampling methods

Although Italy is not a signatory to the Nagoya Protocol, we carried out the sampling with the authorization of our collaborators from the INGV Palermo (Italy), as stated in the acknowledgments.

Sediment samples were collected from shallow hydrothermal marine vents on the shore of Vulcano Island at the Baia di Levante location (GPS coordinates 38°24'45" N, 14°57' 38" E). These sediments and the surrounding fluid were collected in 100 mL Schott bottles filled to the

top and hermetically closed with a rubber stopper (no gas phase). They were stored at 4-8°C in the dark until used for culture.

## 2.2  Isolation of strain V6Fe1$^T$

For enrichment and isolation cultures, the Vulcano High Salt (VHS) medium was used, containing (per liter of distilled water): 18 g NaCl, 1 g MgCl$_2$.6H$_2$O, 1 g MgSO$_4$.7H$_2$O, 1 g NH$_4$Cl, 1 g (NH$_4$)$_2$SO$_4$, 0.3 g KCl, 1 g Na$_2$SO$_4$, 0.1 g CaCl$_2$.2H$_2$O, 0.5 g cysteine-HCl, 3.3 g PIPES, 0.1 g yeast extract, 1 ml K$_2$HPO$_4$/KH$_2$PO$_4$ 1% solution, 1 ml trace mineral element solution [29], 1 ml Se-Ni-W solution (100 ml$^{-1}$: 10 mg Na$_2$SeO$_3$, 20 mg NiCl$_2$.6H$_2$O, 30 mg Na$_2$WO$_4$.2H$_2$O) and 1 ml of 0.1% (w/v) resazurin (Sigma). The pH was adjusted to 6.3 with a 10 M KOH solution. The medium was boiled under a stream of O$_2$-free N$_2$ gas and cooled at room temperature. 50 ml or 5 ml of VHS medium were dispensed into penicillin flasks or Hungate tubes, respectively, degassed under N$_2$/CO$_2$ (80/20, v/v), and subsequently sterilized by autoclaving at 120 °C for 21 min. The Hungate technique [30] was used throughout the study. Sediments at 5% w/v were first used as inoculum in 50 ml flasks supplemented with 13 mM poorly crystalline Fe(OH)$_3$ under 2 bars of H$_2$/CO$_2$ (80/20, v/v), then incubated at 60°C. Microbial growth was observed after three days of incubation and was accompanied by a change in the color of the medium, from rust-colored to dark brown, due to a reduction of ferric ions in ferrous ions. The enrichment culture was subcultured several times under the same growth conditions prior to isolation to obtain pure cultures. The culture was then serially diluted tenfold in the same culture medium and solidified with 0.8% Phytagel (Sigma) (w/v) using the roll tubes technique. The process of serial dilution in roll tubes was repeated three times until the isolates were deemed axenic. One strain, V6Fe1$^T$, was selected for further taxonomic and physiological characterization.

## 2.3  Taxonomic characterization and genome sequencing, assembly, and annotation

Extraction and purification of the genomic DNA from strain V6Fe1$^T$, PCR amplification, and sequencing of the 16S rRNA gene were performed as described previously [31]. The sequence was compared with available sequences in GenBank using a BLAST search [32]. The complete 16S rRNA gene sequence (1538 nucleotides) was later obtained from the genome sequence and deposited in the GenBank database under accession number ON783970. Selected sequences were aligned using Muscle [33], and evolutionary distances were calculated using the Maximum Likelihood method and the General Time Reversible model. Support for internal nodes was assessed by the bootstrap analysis (2000 replicates). These analyses were conducted in MEGA7 [34].

For genome sequencing, high molecular weight DNA was prepared from a 500 mL culture (basal medium containing 20 mM fumarate and 2 g/L yeast extract) cell pellet using a method based on a phenol-chloroform-isoamyl alcohol extraction as previously described [35]. Genomic sequencing was performed at the GenoToul platform (Toulouse, France) by combining long reads technology of Oxford Nanopore to ease assembling (GridION) and high coverage provided by short paired-end reads obtained with Illumina (MiSeq) technology. The methodological details on sequencing, read processing and *de novo* assembly of the genome are provided in Supplementary Methods. Annotation of the genome was performed using the MicroScope platform [36]. The complete genome sequence has been deposited under the accession number CP063375.

Digital hybridization values dDDH were calculated using the Genome-to-Genome Distance Calculator (GGDC) 2.1 (http://ggdc.dsmz.de/). Average nucleotide Identity (ANI) and

Percentage Of Conserved Proteins (POCP) were computed using Protologger v0.9974 (http://www.protologger.de/) [37]. The detection of orthologous proteins shared by strain V6Fe1$^T$, *D. autotrophicus*, *D. desulfuricans,* and *C. nitroreducens* was performed by the OrthoVenn2 web server [38]. Operons, promoters, and terminators predictions were performed by using the FgenesB, BPROM, and FindTerm online tools at (http://www.softberry.com/berry.phtml).

For 16S rRNA gene analysis, the sequences were extracted from genome assemblies using barrnap 0.9 (https://github.com/tseemann/barrnap) and the sequences were aligned with MAFFT 7.515 [39]. The resulting alignment was used for phylogenetic tree inference with IQ-TREE 2.3.4 [40] using the GTR + F + G4 model. Branch support was obtained after 1000 bootstrap replicates.

For phylogenetic analysis based on 120 bacterial marker genes, the sequences were extracted and aligned from genome assemblies using GTDB-Tk 2.3.2 [41] with the GTDB Release 214 and phylogenetic tree inference using IQ-TREE 2.3.4 [40] with LG + I + R4 model. Branch support was obtained after 1000 bootstrap replicates.

## 2.4    Physiological and metabolic characterization

Growth experiments were performed in triplicate. pH, temperature, and NaCl concentration ranges for growth, as well as the oxygen, antibiotics, nitrite (2 mM), and sulfite (2 mM) sensitivities, were determined using the VHS medium supplemented with fumarate (20 mM) and yeast extract (2 g/L) under N$_2$ 100% gas phase. When gas supply was required for metabolism studies (e.g., with H$_2$ or CO tested as electron donors, N$_2$O tested as an electron acceptor, or when oxygen tolerance was tested), cultures were performed in 120 mL penicillin bottles containing 50 mL of medium and incubated at 60°C with shaking at 120 rpm. To determine optimal growth temperature, water baths were used for incubating bacterial cultures from 20°C to 75°C, with increments of 5°C. The pH of the medium was adjusted by injecting in Hungate tubes aliquots of anaerobic stock solutions of 1 M HCl (acidic pHs), 10% NaHCO$_3$, or 8% Na$_2$CO$_3$ (alkaline pHs) to test pH between 4.0 and 8.6 and checked after autoclaving. To study the salinity requirement, NaCl was weighed directly in the tubes (0-10%, w/v) before the medium was dispensed therein. Strain V6Fe1$^T$ was subcultured twice under the same experimental conditions before the growth rates were determined. The culture conditions were then fixed with the optimal temperature, pH, and salinity parameters determined from these experiments.

Susceptibility to antibiotics was tested twice with a final concentration of 100 μg/l of the following antibiotic injected from sterile, anaerobic stock solutions and tested separately: ampicillin, bacitracin, chloramphenicol, cycloheximide, gentamicin, kanamycin, streptomycin, tetracycline, and vancomycin as tested in [26]. The oxygen sensitivity of strain V6Fe1$^T$ was tested at 60°C, up to 12.5% O$_2$.

To determine the substrates used by strain V6Fe1$^T$, VHS medium was used throughout the study. The gas phase consisted of 100% N$_2$. Balch vitamins [29] were added when studying non-proteinaceous substrates except for the autotrophy tests. When testing autotrophic capabilities, cysteine and PIPES were omitted in the medium and the gas phase was H$_2$/CO$_2$ (80/20, v/v 2 bar). Carbon monoxide (CO) in the gas phase (N$_2$/CO 90/10% vol/vol) was tested as an electron donor using nitrate as the electron acceptor [42]. To ensure the capacity of strain V6Fe1$^T$ to reduce nitrous oxide to dinitrogen, strain V6Fe1$^T$ was cultured with nitrous oxide as the sole electron acceptor in the medium, with H$_2$, glucose, or acetate. Gaseous phases were fixed at N$_2$O/H$_2$/CO$_2$ (33/54/13 vol/vol 1 bar) or N$_2$O/N$_2$/CO$_2$ (33/54/13 vol/vol 1 bar) according to [43], for testing nitrous oxide utilization with H$_2$ or acetate (20 mM) respectively as an electron donor. The vials were incubated with shaking at 120 rpm at 60°C.

Sugars (arabinose, cellobiose, fructose, galactose, glucose, N-acetyl glucosamine, glycerol, lactose, maltose, mannose, raffinose, ribose, rhamnose, saccharose, sorbose, xylose), polymers of sugars (starch), alcohols (ethanol, mannitol, methanol), proteinaceous substrates (casamino acids, peptone, tryptone, yeast extract) and organic acids (acetate, butyrate, citrate, crotonate, formate, fumarate, lactate, malate, propionate, pyruvate and succinate) were tested as growth substrates for strain V6Fe1$^T$. Each substrate was added to the VHS medium from sterile anaerobic stock solutions to a final concentration of 20 mM, formate to 100 mM, and starch was added before autoclaving at a final concentration of 10 g/L.

Elemental sulfur (0.5% w/v), sodium sulfate (20 mM), sodium thiosulfate (20 mM), sodium sulfite (2 mM), fumarate (20 mM), sodium nitrate (10 mM), sodium nitrite (2 mM), manganese oxide $MnO_2$ (13 mM) and iron oxide as poorly crystalline $Fe(OH)_3$ (13 mM), goethite (13 mM) or soluble forms such as ferric citrate or Fe(III)-EDTA (13 mM) were tested each as terminal electron acceptors in the presence of the following electron donors tested separately at concentrations already mentioned: arabinose, cellobiose, fructose, galactose, glucose, glycerol, lactose, maltose, mannose, raffinose, ribose, rhamnose, saccharose, sorbose, xylose, starch, ethanol, mannitol, methanol, casamino of acids, peptone, tryptone, yeast extract, acetate, butyrate, citrate, crotonate, formate, fumarate, lactate, malate, propionate, pyruvate and succinate, $H_2/CO_2$ 80/20 v/v at 2 bar as electron donors. When elemental sulfur was provided, autoclaving was performed at 105°C for one hour to prevent sulfur melting. Nitrogen-fixing capabilities were determined in the VHS medium without ammonium supplemented with 20 mM fumarate under $N_2/CO_2$ (80/20, v/v 1 bar). Bacterial growth was monitored by microscopic observation and counting using a Helber Bacteria Counting Chamber at 0.02 mm depth (Hawksley, England).

## 2.5   Cell morphology and chemotaxonomy

Cell morphology and purity of the strains were assessed under an Optiphot (Nikon) phase contrast microscope. For ultrastructure studies, cells were negatively stained with sodium phosphotungstate, as previously described [44] and observed under a Zeiss EM912 microscope. The presence of spores was checked by microscopic observation of cultures and after pasteurization tests performed at 80, 90, and 100°C for 10 and 20 min and in old (more than three weeks) cultures. We tried to stimulate flagella production, providing a remote electron acceptor to the strain. To do that, motility assays were performed in anaerobic tubes by suspending fresh cultures of strain V6Fe1$^T$ in VHS mineral medium with soft agar (0,1 to 1%). After solidification, the cell suspension is covered with sterile VHS medium with soft agar (0,1 to 1%) supplemented with 13 mM Fe(III) and 2 g/L yeast extract under $N_2/CO_2$ (80/20, v/v) or $H_2/CO_2$ (80/20, v/v) at 1 bar each.

Fatty acids were extracted using Miller's modified method [45] and analyzed by gas chromatography (model 6890N, Agilent Technologies) using the Microbial Identification System (MIDI, Sherlock Version 6.1; database, ANAER6). Analyses of the respiratory quinones and polar lipids were conducted at the Identification Service of the DSMZ (Braunschweig, Germany). The G + C content was determined from the genome sequence.

## 2.6   Chemical analyses

Sulfide production was determined *via* the adaptation of a protocol from the LCW 053 Hach kit. Ammonium production was measured spectrophotometrically using the indophenol blue method [46]. Organic compounds were measured by HPLC (Dionex Ultimate 3000). Gas concentrations ($N_2O$, $CO_2$, $H_2$, CO) were measured by gas chromatography (Shimadzu GC 8A). Nitrate, nitrite, sulfate and thiosulfate concentrations were measured by ion chromatography

(Dionex Easion). The converting rates of nitrate to ammonium and/or nitrogen, fumarate to succinate and/or carbon dioxide by whole cells were calculated by comparing the rate of consumption of supplemented substrates and/or formation of the corresponding products.

## 2.7 Investigation of nitrate reduction metabolic pathways

The metabolic fate of $^{15}$N-labeled nitrate was investigated to identify terminal products of nitrate reduction. Culture media containing $^{15}NO_3^-$ 2 mM (Aldrich) were prepared with or without 10 mM unlabeled nitrate as an electron acceptor in the presence of fumarate 15 mM or $H_2/CO_2$ (80/20, v/v) as electron donors. After one week of incubation at 60°C, gaseous dinitrogen and dissolved ammonium isotopic ratios were measured by mass spectrometry (CN-Integra tracer-mass) according to [47]. Briefly, 2 ml of final culture supernatant was treated in a stoppered flask with mild alkali (MgO); conversion of $NH_4^+$ to volatile $NH_3$ was carried out at 60°C for four days. Evolved $NH_3$ was collected as $N-NH_4^+$ in an acidified (coated with 0.5 M $H_2SO_4$) disk cut from a GF/D filter (Whatman) suspended on the stopper. Each disk was rolled into a pellet in the laboratory and fed into a mass spectrometer. An organic standard (glycine) was used to calibrate the measurements and track the consistency of Dumas combustion.

A quantitative RT-PCR approach was used to quantify the expression of several genes involved in nitrate reduction pathways. To determine which genes were expressed according to the electron donor furnished during nitrate reduction, strain V6Fe1$^T$ was cultivated in three conditions: condition F with fumarate 15 mM as the only substrate (reference condition for gene expression), condition FN with 15 mM fumarate and 10 mM nitrate and condition HN with $H_2/CO_2$ (80/20, v/v) and 10 mM nitrate (autotrophic condition). Cultures of 100 mL in 300 mL glass flasks, hermetically closed with a rubber stopper and aluminum seal, were incubated at 60°C, with shaking at 120 rpm. Aliquots of 2 mL were regularly sampled for chemical analyses and cell counting. When cells reached the mid-exponential phase, 20 mL of culture were sampled, centrifuged, and immediately stored at -80°C for later RNA extraction.

Specific primers for the selected genes (Sup. Table 1) were designed using Primer3 software version 4.1.0 and ordered to Eurogentec (Liège, Belgium). Automatic genome annotation allowed us to identify the following genes: *cym*A DSN97_06310, *hcp* DSN97_01195, *napA* DSN97_05245, *nor*V DSN97_07985, *nrfD* (present in two adjacent reversed copies DSN97_01825 and DSN97_11315). They encode respectively a cytoplasmic membrane electron transport protein involved in nitrate reduction by *Shewanella oneidensis* [48], a hydroxylamine reductase, a periplasmic nitrate reductase, a nitric oxide reductase and an integral transmembrane protein involved in the terminal transfer of electrons from the quinone pool into the terminal components of the Nrf pathway. A blast search within the V6Fe1$^T$ genome revealed the presence of a putative *nrfA* gene DSN97_03745, encoding for a pentaheme nitrite reductase, putatively involved in DNRA. Primers of 20 to 24 bp have been designed to generate amplicons of 152-208 bp length and a melting temperature between 60 and 62°C (Sup. Table 1), except primers targeting the 16S rRNA gene DGGE300F1 [49] and univ5161 [50], already available at the lab, whose sequence differed only by one base from those of strain V6Fe1$^T$.

RNA extractions were performed with the High Pure RNA Isolation Kit (Roche). RNA samples were then treated with the TurboDNaseTM (Ambion®, Thermo Fisher Scientific Corp.) according to the manufacturer's recommendations and reverse transcribed into cDNA using the SuperScript® IV Reverse Transcriptase (Invitrogen). The relative gene expression levels were quantified using SYBR Green on a CFX Real-Time PCR system (Biorad). Quantitative PCR was performed with 2 µL cDNA (directly reverse transcribed DNA and $10^{-1}/10^{-2}$ dilutions), 0.25 µL of each 10 µM primer, 10 µL of 2X SyberGreen and 7.5 µL of ultrapure

**Table 1. Characteristics that differentiate strain V6Fe1$^T$ from its closest relatives.** Strains: 1, strain V6Fe1$^T$; 2, *Deferrivibrio essentukiensis* Es71-Z0220$^T$ [52], 3, *Petrothermobacter organovorans* ANA$^T$ [26]; 4, *Calditerrivibrio nitroreducens* YU37-1$^T$ [53]; 5, *D. desulfurricans* SSM1$^T$ [25]; 6, *D. abyssi* [28]; 7, *D. autotrophicus* SL50$^T$ [27].

| Characteristics | 1 | 2 | 3 | 4 | 5 | 6 | 7 |
|---|---|---|---|---|---|---|---|
| Temperature range (°C) | 45–65 | 25–54 | 25–60 | 30–65 | 40–70 | 45–65 | 25–75 |
| Optimum (°C) | 60 | (45–50) | 55 | 55 | 60–65 | 60 | 60 |
| NaCl range (w/v, %) | 0–8 | 0–1.8 | 0–6 | 0–0.5 | 1.8–9.6 | 1–5 | 1–6 |
| Optimum (w/v, %) | 2.5 | 0.7 | 2.5 | 0.15 | 3.6 | 3 | 2.5 |
| pH range | 5.1–7.7 | 6–8.2 | 6–8 | 5.5–8 | 5–7.5 | 6–7.2 | 5–7.5 |
| Optimum | 6.3 | 7.1 | 7 | 7–7.5 | 6.5 | 6.5 | 6.5 |
| Fermentation | + | – | + | – | + | – | – |
| Electron donors | | | | | | | |
| Ethanol | – | – | – | – | + | – | – |
| Formate | + | – | ND | – | + | – | + |
| Propionate | – | ND | – | – | + | – | + |
| Fumarate | + | + | + | + | + | ND | – |
| Electron acceptors | | | | | | | |
| Elemental sulfur | + | + | ND | – | + | + | + |
| Fe(III) | + | + | + | – | – | + | + |
| Mn(IV) | + | + | + | – | – | – | + |
| Sulfate | – | – | + | – | – | – | – |
| Thiosulfate | – | – | ND | – | – | – | – |
| Nitrate | + | + | + | + | + | + | + |
| Tolerance to O$_2$ | – | – | – | ND | – | – | – |
| End product of nitrate Reduction | N$_2$/ NH$_4^+$ | NH$_4^+$ | NO$_2^-$ | NH$_4^+$ | NO$_2^-$ | NO$_2^-$ | NH$_4^+$ |
| DNA G + C content (mol%) | 34.8 | 34.04 | 34.3 | 35.1 | 38.6 | 30.8 | 28.7 |

+, Positive; –, negative.

RNAse/DNAse-Free distilled water (Thermofisher). Thermal cycling conditions were as follows: 2 min at 98°C, followed by 30 cycles of 5 s at 98°C, 30 s at 61°C, and 20 s at 72°C. Data collection was performed during each annealing phase. Expression levels of all the genes were normalized to 16S rRNA gene expression. Gene expression was normalized against the 16S rRNA gene using the $2^{-\Delta CT}$ calculation: $\Delta C_T = C_{T\ gene\ of\ interest} - C_{T16S\ rRNA}$ ($C_T$, threshold cycle).

## 3   Results and discussion

### 3.1   Metabolic properties and taxonomic characterization of strain V6Fe1$^T$

Environmental samples were collected from shallow hydrothermal sediments in Baia di Levante, Vulcano Island, Italy. These samples were hydrothermally heated sediments, debris flow deposits, and poorly sorted coarse sands to fine leucite gravels corresponding to the high alkaline character of the lavas [51]. The fluid temperature at the emission point was 55-65°C and pH 6.25. Enrichment cultures were performed from these sediments, with an H$_2$/CO$_2$ (80/20, v/v) gas phase, Fe(OH)$_3$ 13 mM as the electron acceptor, and 0.1 g/L yeast extract. Brown coloration within three days indicated iron reduction. Isolation of axenic cultures was performed using the Hungate roll-tube method. Several black colonies with diameters ranging from 1.0 to 2.0 mm developed after 3-5 days of incubation at 60°C. They were picked

separately, and the isolation procedure was repeated twice. Finally, one axenic culture was named strain V6Fe1[T] and characterized. Based on its 16S rRNA gene sequence, the most closely related type species are *Deferrivibrio essentukiensis* Es71-Z0220[T] (99.5% 16S rRNA gene identity) [52], *P. organivorans* ANA[T] (94.2% identity) [26], *C. nitroreducens* YU37-1[T] (89.4% identity) [53] and *D. desulfuricans* SSM1[T] (88.8% identity) [25]. A comparison of 1538 nucleotides of the 16S rRNA gene sequence of strain V6Fe1[T] with those available in the GenBank database was used to infer a phylogenetic tree (Fig 1). A genomic tree (Fig 2) was also inferred from 120 bacterial marker genes by GTDB-Tk, showing an equivalent position of *Deferrivibrio metallireducens* V6Fe1[T] in the family *Deferrivibrionaceae*, related to the families *Deferribacteraceae* and *Geovibrionaceae*, as well as the cladogram presented on supplementary Fig S1.

Cells of strain V6Fe1[T] are non-motile and produce thin (0.2-0.8 μm x 4-8 μm) slightly curved rods that stain Gram-negative (Gram staining reaction). Indeed, ultrathin sections of cells observed by MET confirmed a Gram-negative type of cell wall (Fig. 3). No spore formation was observed in any tested condition, as is the case with all members of the *Deferribacterota* phylum. No flagella or motility were observed, whereas 40 genes involved in the flagellum synthesis (e.g., 8 genes in an operon DSN97_01525-60) and 12 genes involved in chemotaxis (e.g., CheR DSN97_07015, CheV DSN97_00765, CheW DSN97_06610, CheY DSN97_07060, and CheZ DSN97_07055) are found in the genome. Results in motility tests and flagella observation are contrasted within the *Deferribacterota* phylum. *D. essentukiensis* [52], *C. nitroreducens* [53], *D. autotrophicus* [27], and *D. abyssi* [28], exhibit motility with a polar flagellum; *D. desulfuricans* [25] and *D. thermophilus* [54] exhibit a polar flagellum but no motility whereas *P. organivorans* exhibit no polar flagellum nor motility.

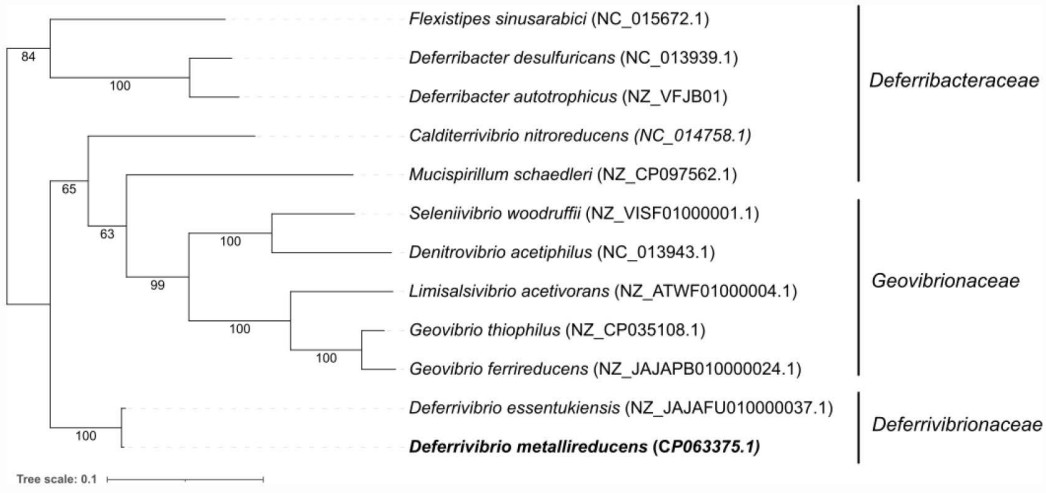

**Fig 1. Phylogenetic tree based on 16S rRNA gene sequences showing the position of *Deferrivibrio metallireducens* V6Fe1[T] in the family *Deferrivibrionaceae*, related to the families *Deferribacteraceae* and *Geovibrionaceae*.** The outgroup (not represented here) was composed of the following type species: *Thermanaeromonas toyohensis* ToBE, *Moorella thermoacetica* DSM 2955, *Moorella humiferrea* OCP, *Moorella glycerini* DSM 11254, *Desulfonatronum thiosulfatophilum* ASO4-2, *Desulfonatronum thioautotrophicum* ASO4-1, *Desulfonatronum thiodismutans* MLF1, *Desulfonatronum lacustre* DSM 10312, *Desulfovibrio ferrophilus* IS5, *Desulfovibrio subterraneus* ND17, *Desulfovibrio mangrove* FT415, *Desulfovibrio desulfuricans* L4, *Desulfovibrio fairfieldensis* CCUG 45958, *Candidatus Desulfovibrio trichonymphae* Rs-N31. 16S rRNA gene sequences were extracted from genome assemblies using barrnap 0.9, and the multiple alignments of the sequences were performed with mafft 7.515 [39]. The maximum-likelihood phylogenetic tree was generated using IQ-TREE 2.3.4 [40] with GTR + F + G4 model. Bootstraps are depicted at nodes, estimated by IQ-TREE -B 1000 option. The tree was generated using iTOL v6, providing a visual representation of the phylogenetic connections. The scale bar indicates the number of substitutions per site.

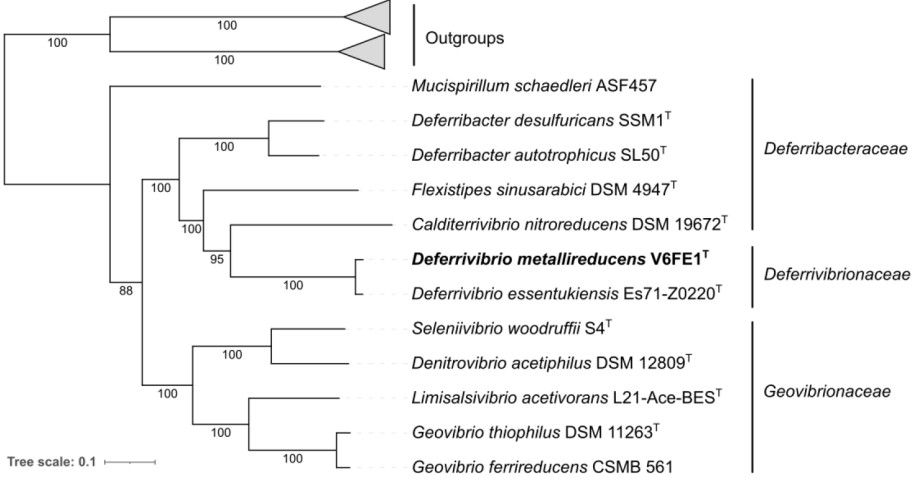

**Fig 2. Phylogenetic tree based on 120 bacterial marker genes by GTDB-Tk showing the position of *Deferrivibrio metallireducens* V6FE1ᵀ in the family *Deferrivibrionaceae,* related to the families *Deferribacteraceae* and *Geovibrionaceae.*** The outgroup (not represented here) was composed of the following type species: *Thermanaeromonas toyohensis* ToBE, *Moorella thermoacetica* DSM 2955, *Moorella humiferrea* OCP, *Moorella glycerini* DSM 11254, *Desulfonatronum thiosulfatophilum* ASO4-2, *Desulfonatronum thioautotrophicum* ASO4-1, *Desulfonatronum thiodismutans* MLF1, *Desulfonatronum lacustre* DSM 10312, *Desulfovibrio ferrophilus* IS5, *Desulfovibrio subterraneus* ND17, *Desulfovibrio mangrove* FT415, *Desulfovibrio desulfuricans* L4, *Desulfovibrio fairfieldensis* CCUG 45958, *Candidatus Desulfovibrio trichonymphae* Rs-N31. 120 bacterial marker genes were extracted and aligned from genome assemblies using GTDB-Tk 2.3.2 [41] with the GTDB Release 214. The maximum-likelihood phylogenetic tree was generated using IQ-TREE 2.3.4 [40] with LG + I + R4 model. Bootstraps are depicted at nodes, estimated by IQ-TREE (-B 1000 option). The tree was generated using iTOL v6, providing a visual representation of the phylogenetic connections. The scale bar indicates the number of substitutions per site.

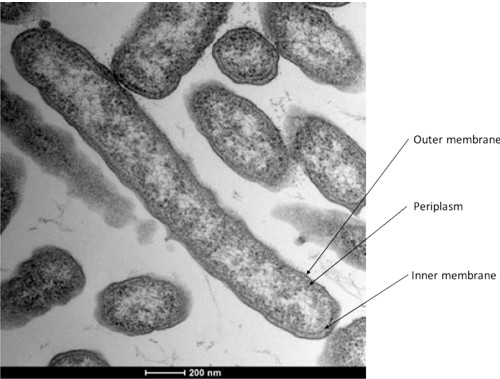

**Fig 3. Transmission electron microscopy of strain V6Fe1ᵀ grown on fumarate 20 mM and yeast extract 2 g/L showing a Gram-negative structure.** Average width 0.2-0.4 μm, average length 4-8 μm. This picture was selected from the 25 pictures we took.

Analysis of the lipid composition of strain V6Fe1ᵀ membranes showed that the major fatty acids are anteiso-$C_{15:0}$ (22.2%), iso-$C_{14:0}$ (13.1%), $C_{16:0}$ (6.8%), iso-$C_{17:0}$ (6.8%), anteiso-$C_{17:0}$ (6.3%), iso-$C_{16:0}$ (6.1%), anteiso-$C_{17:1}$ (6.1%), iso-$C_{17:1}$ (5.9%), $C_{18:0}$ (3.7%), $C_{17:0}$ (2.6%), iso-$C_{16:1}$ (2.0%), $C_{16:1}$ (1.7%). The major fatty acids of strain V6Fe1ᵀ differ from those of *D. essentukiensis*. In strain V6Fe1ᵀ, unsaturated fatty acids account for more than 15%, but only 4% in *D. essentukiensis*. This discrepancy probably reflects the different salinities these strains face in

their environments [55]. Strain V6Fe1$^T$ inhabits a hydrothermal heated seawater environment, whereas *D. essentukiensis* inhabits mineral water deposits with low salinity [52], as *C. nitroreducens,* in whose cell wall no unsaturated fatty acid was detected, does [53]. Indeed, unsaturated fatty acids of *Limisalsivibrio acetivorans* L21-Ace-BES$^T$, a recently discovered bacterium belonging to the *Deferribacteres* class, isolated from a hypersaline lake, account for 48,7% of the total fatty acid content [56]. Major respiratory quinones are menaquinone MK-7 40% and MK-8 60%. MK-7 is the principal cofactor of CymA, a quinone-converting enzyme that plays a central role in the multibranched respiratory chain of *S. oneidensis* [57]. The polar lipid profile of strain V6Fe1$^T$ consisted of glycolipids, phospholipids, phosphatidylglycerol, phosphatidylethanolamine, and diphosphatidylglycerol (supplementary Fig S2).

The phenotypic characteristics of strain V6Fe1$^T$ are listed in Table 1. It grows at temperatures between 45°C and 65°C, with an optimum at 60°C (no growth at 40°C and 70°C). Growth occurred at NaCl concentrations between 0 and 80 g/L, with an optimum at 25 g/L (no growth at 90 g/L). Growth occurred between pH 5.1 and 7.7, with optimal growth at pH 6.3–6.5. In optimal fermentative conditions, e.g., with 20 mM fumarate and 2 g/L yeast extract, the doubling time was measured at 120 minutes. Strain V6Fe1$^T$ is resistant to streptomycin and vancomycin, while it is sensitive to ampicillin, bacitracin, chloramphenicol, cycloheximide, gentamicin, kanamycin, and tetracycline. Nitrite and sulfite (both at 2 mM) inhibited growth. Strain V6Fe1$^T$ showed an oxygen tolerance of up to 0.6% in the gas phase, while no growth was observed with 1.5% oxygen in the gas phase, as is the case for *D. thermophilus* [54]. In comparison, *D. essentukiensis* and *C. nitroreducens* do not tolerate 2% oxygen in the gas phase and were described as obligate anaerobes [52,53]. Strain V6Fe1$^T$ can use the following organic acids and proteinaceous substrates as carbon and energy sources: butyrate, fumarate, lactate, malate, pyruvate, succinate, yeast extract, peptone, tryptone, casamino acids. Sugars (arabinose, cellobiose, fructose, galactose, glucose, N-acetyl glucosamine, lactose, mannose, rhamnose, ribose, saccharose, starch, xylose), propionate, citrate, methanol, ethanol, formate, and $H_2/CO_2$ (80/20, v/v) in the presence of acetate (2 mM acetate) were not used.

Fermentation is a rare trait among members of the *Deferribactota* phylum and has only been observed in *P. organivorans* [26] and *D. desulfuricans* [25]. *D. essentukiensis*, the closest relative of strain V6Fe1$^T$, cannot ferment any substrate [52], like *C. nitroreducens* [53]. Sugar utilization within members of the *Deferribacterota* phylum has only been reported for *D. autotrophicus* which can use maltose as a carbon and energy source [27]. Strain V6Fe1$^T$ can grow by anaerobic respirations, using organic acids or proteinaceous substrates as electron donors, with Fe(III), Mn(IV), S° and $NO_3^-$ as electron acceptor but not sulfate nor thiosulfate or soluble Fe(III) forms. Strain V6Fe1$^T$ could not use carbon monoxide as an electron donor as *D. autotrophicus* [42]. Yeast extract was not necessary for the growth of strain V6Fe1$^T$ but strongly stimulated it.

Autotrophic growth was studied in a minimal VHS medium without organic carbon (i.e., without yeast extract, Balch vitamins, and PIPES salt). Strain V6Fe1$^T$ was able to grow under autotrophic conditions using $CO_2$ as the sole carbon source, $H_2$ as the sole electron donor and $NO_3^-$, $N_2O$, Fe(III), Mn(IV), or S° as electron acceptors.

Strain V6Fe1$^T$ shared many phenotypic features with representatives of the genera *Deferrivibrio, Petrothermobacter, Calditerrivibrio,* and *Deferribacter* (Table 1). Species of these genera were mainly isolated from hydrothermal vents or terrestrial hot springs. In contrast, *D. essentukiensis* [52] was isolated from a subsurface aquifer, and *D. thermophilus* [54] and *P. organivorans* [26] were isolated from oil reservoirs. All these strains have a Gram-negative type of cell wall structure. Members of the *Deferribacterales* order (*Deferrivibrio, Deferribacter, Calditerrivibrio,* and *Petrothermobacter* species) are rod-shaped and grow by anaerobic respiration (using a variety of complex organic compounds and organic acids with diverse electron

acceptors), but only a few can ferment. On the contrary, strain V6Fe1$^T$ and its closest relatives, *P. organivorans* and *D. desulfuricans* can grow by fermentation of proteinaceous compounds and organic acids. Within the *Deferribacterota* phylum, only *D. autotrophicus* [27], *D. abyssi* [28], and strain V6Fe1$^T$ can grow chemolithoautotrophically using $CO_2$ as the sole carbon source and $H_2$ as the electron donor.

## 3.2  General genome properties

The genome of strain V6Fe1$^T$ consists of a circular chromosome with an overall size of 2,358,333 bp and a G + C content of 34,83 mol %, like that of closely related strains with sequenced genomes (2,382,387 bp for *D. essentukiensis*, 2,216,552 bp for *C. nitroreducens* and 2,542,993 bp for *D. desulfuricans*). General features of strain V6Fe1$^T$ genome are listed in supplementary Table S2. The genome contains 2356 coding sequences (CDS), of which 2254 are protein coding. Of the 2254 protein-coding genes, 1937 were given a predicted function. Signal transduction mechanisms, ribosome structure and biosynthesis, membrane and envelope biosynthesis, and cell motility were among the most represented functional categories with respectively 7.5, 6.3, 5.8, and 4.2% of the function-predicted genes. Strain V6Fe1$^T$, *D. essentukiensis, D. desulfuricans, D. autotrophicus,* and *C. nitroreducens* are close relatives (Fig 1). *D. desulfuricans, D. autotrophicus* and *C. nitroreducens* share 1696, 1702 and 1594 proteins, respectively within 1922 strain V6Fe1$^T$ proteins detected by the Orthovenn2 web server (Supplementary Fig S3).

The recommended threshold value for a new species is around 95-96% ANI [58]. Measurement of ANI values showed that strain V6Fe1$^T$ shared 89.79%, 76.91%, and 76.89% ANI with *D. essentukiensis*, *D. desulfuricans,* and *C. nitroreducens*, respectively, indicating that strain V6Fe1$^T$ belongs to a new species. The dDDH values between the genomes of strains V6Fe1$^T$ and *D. essentukiensis* was 77.8%, higher than the 70% limit [58], while they were much lower with *D. desulfuricans, C. nitroreducens, Flexistipes sinusarabicus,* and *D. autotrophicus*, respectively 13.3%, 13.0%, 12.8%, and 12.6%. Percentage of Conserved Proteins (POCP) calculated with Protologger indicated values of 68.05% with *D. desulfuricans,* 67.32% with *C. nitroreducens,* 63.34% with *F. sinusarabici,* 47.89% with *Denitrovibrio acetiphilus* and 38.55% with *Mucispirillum schaedleri.* A 50% POPC value threshold was recommended to define a new genus [59]. Such a recommendation should lead to transferring and reclassifying all *Deferrivibrio, Deferribacter, Calditerrivibrio, Flexistipes, and Petrothermobacter* species into the same genus. Classification of strain V6Fe1$^T$ as a new species of the genus *Deferrivibrio* is proposed within the *Deferrivibrionaceae* family.

## 3.3  Central carbon metabolism

The genome of strain V6Fe1$^T$ encodes glycoside hydrolases (of which one α-amylase), proteases and peptidases, alcohol and aldehyde dehydrogenases, and genes coding for ABC-transporters of sugars and peptides. Whereas sugars were not used by strain V6Fe1$^T$ during our experiments, searches against the carbohydrates-active enzymes (CAZYmes) database [60] identified 47 CAZymes, including 11 glycoside hydrolase, 29 glycoside transferase, 2 carbohydrate esterase, and 5 carbohydrate-binding enzyme genes, probably involved in the peptidoglycan metabolism. Sugars are generally not used by members of the *Deferribacterota,* except for *D. autotrophicus* [27]. Gluconeogenesis pathways, pentose phosphate pathway (only non-oxidative phase), and a complete Embden-Meyerhoff pathway were identified. The genes for the complete biosynthesis pathways of many amino acids are present: alanine, arginine, aspartate, cysteine, glutamate, glutamine, glycine, phenylalanine, threonine, and tyrosine. The genome also encodes five dipeptides and at least 18 amino-acids ABC transporters with

specificity for polar amino acids and branched amino acids, confirming amino acids as substrates for growth.

The presence of several cytochromes and ATP synthase subunits, as well as menaquinones genes, affords strain V6Fe1$^T$'s respiratory capacities. This strain harbors a complete set of enzymes involved in the Krebs cycle. Oxidative phosphorylation in strain V6Fe1T is performed by a typical F0F1-ATP Synthase, whose subunits are encoded by two distinct operons (DSN97_00830-860 and DSN97_06300-305).

Strain V6Fe1$^T$ grows chemolithoautotrophically using $CO_2$ as the carbon source, $H_2$ as an electron donor, and Fe(III), Mn(IV), S°, $NO_3^-$ or $N_2O$ as electron acceptors. Key enzymes of classical carbon fixation pathways (Calvin-Benson cycle, reductive acetyl-CoA pathway, reductive tricarboxylic acid cycle, 3-hydroxypropionate/4-hydroxybutyrate cycle, dicarboxylate/4-hydroxybutyrate cycles and 3-hydroxypropionate bi-cycle) were not found within the genome. The capacity for autotrophic growth was experimentally shown for *D. autotrophicus* [27] and *D. abyssi* [28] and suggested for *Geovibrio thiophilus* [61]. The genome of strain V6Fe1$^T$ contains the complete set of TCA genes cycle, like those of *D. autotrophicus* and *G. thiophilus*. Recently, a new carbon fixation pathway has been proposed for the sulfur-reducing anaerobic deltaproteobacterium *Desulfurella acetivorans* [62]. Under chemolithoautotrophic conditions, citrate synthase can cleave citrate adenosine triphosphate into acetyl coenzyme A and oxaloacetate. Acetyl-CoA is then reductively carboxylated to pyruvate by ferredoxin-dependent pyruvate synthase. There, the TCA cycle operates in the reductive direction with the reverse reaction of citrate synthase and this new pathway is called the 'reversed oxidative TCA cycle' (roTCA) [62]. Based on our experimental and genomic results, we assume strain V6Fe1$^T$ also uses the roTCA cycle for carbon fixation under chemolithoautotrophic conditions.

The roTCA cycle is indeed challenging to infer from genomic data. *D. essentukiensis, C. nitroreducens,* and *D. desulfuricans* possess all the genes necessary for roTCA carbon fixation (like *D. autotrophicus* and strain V6Fe1$^T$) but could not grow autotrophically [25,52,53]. Autotrophy by roTCA is not thermodynamically feasible in most environmental conditions but is favored by high partial $CO_2$ concentration (p$CO_2$), as was shown in *Hippea maritima* cultures [63]. High $CO_2$ fluxes are usually measured at hydrothermal vents [64], especially in the Vulcano system where the p$CO_2 > 98\%$ in the hydrothermal fluid [2]. Autotrophy via roTCA could thus be an adaptive advantage for strain V6Fe1$^T$ in this environment.

### 3.4   Sulfur and iron respiratory genes

Strain V6Fe1$^T$ can reduce elemental sulfur (S°) to sulfide, oxidizing various organic compounds or $H_2$ (Table 1). Sulfate or thiosulfate were not used as electron acceptors. Strain V6Fe1$^T$ is unable to perform sulfide oxidation. Among the closest related strains, *D. essentukiensis, D. desulfuricans*, and *D. autotrophicus* reduce S° into sulfide, while *C. nitroreducens* cannot. This ability has not been tested for *P. organivorans*, which is unique to the *Deferribacteraceae* for its sulfate reduction capacity.

Two operons are plausible candidates for sulfur reduction. The genome encodes a duplicated operon containing NrfD polysulfide reductases (DSN97_01825 and DSN97_11315), the two gene copies being repressed when nitrate is supplied in the medium (see section 3.5). These operons are also present in *D. essentukiensis, D. autotrophicus,* and *D. desulfuricans* genomes, while *C. nitroreducens* has only one *nrf*D gene (CALNI_RS00910) (Fig 4). In another operon, the gene coding the subunit B of a molybdopterin tetrathionate reductase (Ttr, DSN97_01830), an enzyme that reduces tetrathionate to thiosulfate, is associated with molybdopterin oxidoreductase subunits (DSN97_01830). Like in the *D. desulfuricans* genome, this operon is adjacent to another operon containing an outer membrane phosphate selective porin DSN97_01845 and a rhodanese domain-containing protein DSN97_01850. These

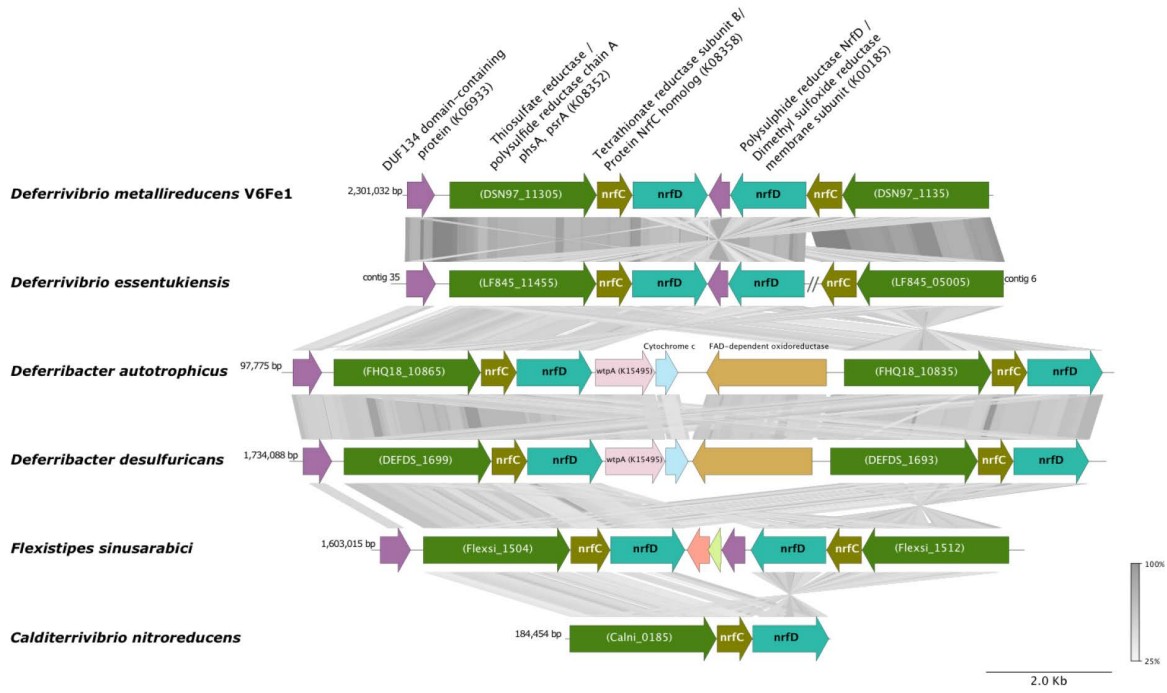

**Fig 4. Comparative genome analysis of NrfD complex region among *Deferrivibrio metallireducens* V6Fe1^T and its closest relatives.**
CDS are represented by arrows indicating their orientation (direct or complementary strand) in the genome. CDS numbers are shown below the corresponding arrow according to the annotation by the CDS, which at MaGe (https://mage.genoscope.cns.fr/). The figure was generated using pyGenomeViz v1.0.0. The degree of amino acid similarity between paired ORFs was analyzed by tBLASTx search, and the percentage of similarity is represented in color from pale gray to dark gray. Note that the duplication plus inversion concerns Strain V6Fe1^T, *D. essentukiensis*, and *F. sinusarabici* genomes; *D. desulfuricans* and *D. autotrophicus* present a duplication without inversion, *C. nitroreducens* present no duplication.

two proteins may facilitate sulfur respiration in strain V6Fe1^T, acting as a polysulfide-specific porin in the outer membrane and a polysulfide-binding protein in the periplasmic space, respectively.

Strain V6Fe1^T can reduce Fe(III) insoluble minerals but not soluble forms. In the *Deferribacterota* phylum, *D. essentukiensis* and *D. autotrophicus* are the only members capable of iron reduction whose genomes have been sequenced [28,53]. Strain V6Fe1^T genome contains 17 genes encoding putative multiheme c-type cytochromes and two 'e-pilin' genes (two type IV pilins genes PilA DSN97_09015 and DSN97_02165), some of which are probably involved in Fe(III) reduction [65]. We found the gene coding for the quinone-converting enzyme CymA (DSN97_06310). Its specific cofactor, the menaquinone 7 MK-7 [57], was detected by biochemical methods (see 3.1). CymA is a monotopic membrane tetraheme c-type cytochrome belonging to the NapC/NirT family and central to anaerobic respiration in *Shewanella* sp. CymA could be crucial in shutting electrons to Fe(III) outside the cell *via* a small tetraheme cytochrome (STC) as a periplasmic shuttle [66]. A cytochrome c (DSN97_05235) closely related to the octaheme cytochrome of *D. autotrophicus* (FHQ18_RS08740) and the octaheme cytochrome of *D. essentukiensis* (LF845_RS02975) was found. This cytochrome is up-regulated in the presence of ferrihydrite in a thermophilic Gram-positive bacterium [67] and has been proposed to be a key determinant of iron reduction in *Deferribacterales*, since it is present in members of *Deferribacterales* that reduce iron, whereas it is absent in *D. desulfuricans* and *C. nitroreducens*, which are incapable of iron reduction.

### 3.5 Nitrogen metabolism and metabolic pathways involved in nitrate reduction

Strain V6Fe1T can use nitrate as an electron acceptor with numerous electron donors, as can all of its close relatives (Table 1). In the strain V6Fe1$^T$ genome, we found the periplasmic nitrate reductase *nap*A gene DSN97_05245 in an operon with genes coding a multiheme cytochrome c, adjacent to another operon containing the genes of chaperone protein *nap*D, *nap*G, and those coding a ferredoxin protein from the NapH/MauN family (Fig 5). This gene organization is in synteny with the same gene clusters found in the genomes of *D. essentukiensis, D. autotrophicus,* and *D. desulfuricans*. NapG and NapH subunits are assumed to form an energy-conserving quinol dehydrogenase [68]. The gene for the membrane-bound NarG nitrate reductase was not found in the genome. As NapA has a higher affinity for nitrate than the membrane-bound enzyme NarG [69], its occurrence in strain V6Fe1$^T$ may represent an adaptation to the low nitrate concentrations typically found in vent fluids [12].

The genome of strain V6Fe1T contains neither assimilatory (*nir*A, *nir*B) nor dissimilatory (*nir*K) nitrite reductase genes. Nevertheless, a formate-dependent nitrite reductase gene *nrf*A (DSN97_03745) and a *nrf*D gene (present in two adjacent reversed copies, DSN97_01825 and DSN97_11315) were identified, possibly involved in the respiratory ammonification [48]. No nitric oxide reductase (*nos*Z) was found in the genome. The hem-containing (cd1-NIR, *nir*S) and the copper-containing (Cu-NIR, *nir*K) nitrite reductases genes catalyzing the reduction of nitrite ($NO_2^-$) to nitric oxide (NO) in most nitrate-reducing bacteria were absent in the V6Fe1$^T$ genome. However, the Hcp encoded by *hcp* (hybrid cluster protein) DSN97_01195 could reduce nitric oxide to nitrous oxide ($N_2O$) or hydroxylamine to ammonium. A gene encoding a subunit of the anaerobic nitric oxide reductase flavorubredoxin NorV DSN97_07985 was found, which could also be involved in reducing nitric oxide to nitrous oxide. No enzyme associated with the oxidative part of the nitrogen cycle, nor any enzyme involved in nitrogen fixation (except a NifU family protein, DSN97_10775) was identified in the genome.

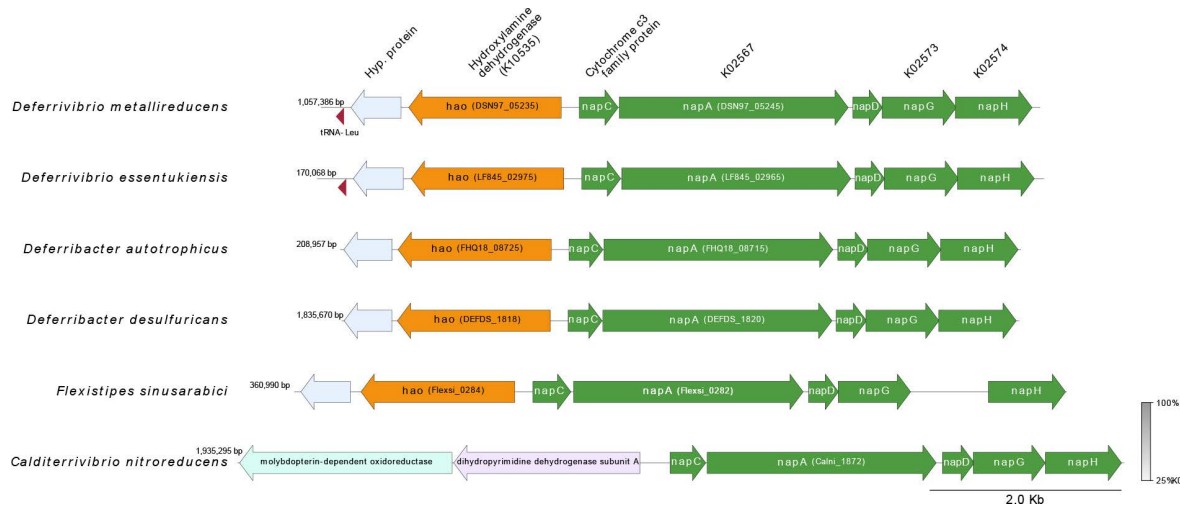

**Fig 5. Comparative genome analysis of the region encoding the NapA complex genes of *Deferrivibrio metallireducens* V6Fe1$^T$ and its closest relatives.** CDS are represented by arrows indicating their orientation (direct or complementary strand) in the genome. CDS numbers are shown below the corresponding arrow according to the annotation by MaGe (https://mage.genoscope.cns.fr/). The figure was generated using pyGenomeViz v1.0.0. The degree of amino acid similarity between paired ORFs was analyzed by tBLASTx search, and the percentage of similarity is represented in color from pale gray to dark gray.

Cultures with $^{15}$N-labelled nitrate were performed to monitor the fate of nitrate. The standard isotopic ratio of atmospheric nitrogen is 0.366 ± 0.001 [70]. Heavier ratios showed enrichment of the heavy $^{15}$N isotope, indicating the fate of $^{15}$N-labelled nitrate (Table 2). Strong enrichment of both $N_2$ and $NH_3$ molecules was frequently observed, which proves the formation of isotopically heavy dinitrogen gas and ammonium from nitrate. Culture experiments in which nitrate was limited (2 mM $^{15}$N nitrate with 20 mM fumarate) showed enrichment in $NH_4^+$ ($^{15}N/^{14}N$ = 0.516) but not in $N_2$ ($^{15}N/^{14}N$ = 0.365). When fumarate and nitrate were furnished at comparable concentrations (20 mM vs. 22 mM), $^{15}$N enrichment was observed in $N_2$ and $NH_4^+$ pools ($^{15}N/^{14}N$ = 0.492 and 1.36, respectively). Culture in autotrophic conditions showed less pronounced but significant enrichments, regardless of the $NO_3^-$ concentrations. $^{15}N/^{14}N$ ratios in $N_2$ and $NH_4^+$ were between 0.377 and 0.372 and between 0.407 and 0.443 with 2 mM or 20 mM $NO_3^-$. Dinitrogen and ammonium isotopic ratios were linked to the concentration and nature of electron donors (fumarate or hydrogen) and acceptor (nitrate). These results show that strain V6Fe1$^T$ can use nitrate reduction pathways, denitrification, and DNRA. The relative importance of these pathways seems to depend on the electron donor's nature and concentration.

New culture experiments were performed with fumarate 15 mM (condition FN) or dihydrogen (in $H_2/CO_2$, 80/20, v/v 1 bar) (condition HN) and nitrate 10 mM as the electron acceptor. Monitoring the concentrations in nitrite, nitrate, dihydrogen, carbon dioxide, nitrous oxide, fumarate, and succinate was used to specify the stoichiometry of the metabolic reactions and then identify the metabolic pathway involved in nitrate reduction (Fig 6). For comparison, strain V6Fe1$^T$ was cultivated in parallel in fermentative conditions (F) with only fumarate 15 mM, used as the reference conditions in RT-qPCR analyses. Cells grew well in all conditions, entering the stationary phase at densities up to $1{,}8.10^9$ (F condition), $3{,}4.10^8$ cells. mL$^{-1}$ (FN conditions) and $1{,}5.10^8$ cells. mL$^{-1}$ (HN conditions) (Fig 6). Differences between cell yield in fermentative and respiratory conditions may be due to the toxicity of nitrite accumulating in the medium (see section 3.1).

In the reference condition (F condition), succinate was produced while fumarate concentrations decreased. The stoichiometry of the reaction was about 2 moles of succinate formed per 3 moles of fumarate consumed. It agrees with previous results acquired on *Clostridium formicoaceticum* [71]: 3 fumarate + 2 $H_2O$ → 2 succinate + 1 acetate + 2 $CO_2$. For the fumarate plus nitrate (FN) condition, the theoretical denitrification reaction was 5 $C_4H_4O_4$ + 12 $NO_3^-$ + 12 $H^+$ → 20 $CO_2$ + 6 $N_2$ + 16 $H_2O$. According to the above equation, 2.07 to 2.35 moles of nitrate were consumed per mole of fumarate in the exponential phase, close to the theoretical stoichiometric ratio value of 2.4. Nitrate was almost wholly consumed, dropping from 8 mM to 0.4 mM. Since no ammonium production was observed (the ammonium concentration remained stable), nitrate reduction was exclusively attributed to the respiratory denitrification of $N_2$. In the hydrogen plus nitrate (HN) condition, no $N_2O$ (denitrification

**Table 2. The isotopic ratio of dinitrogen and ammonium in the gas phase measured after one week of culture spiked with 2 mM $^{15}NO_3^-$.**

| Sample | ($^{15}N/^{14}N$) of dinitrogen in the gas phase % | ($^{15}N/^{14}N$) of ammonium in the liquid phase % |
|---|---|---|
| Glycocolle (reference) | 0,367 | 0,367 |
| Fumarate + 2 mM $^{15}NO_3^-$ | 0,365 | 0,516 |
| Fumarate + 2 mM $^{15}NO_3^-$ + 20 mM $NO_3^-$ | 0,492 | 1,360 |
| $H_2/CO_2$ 80:20 1 bar + 2 mM $^{15}NO_3^-$ | 0,377 | 0,407 |
| $H_2/CO_2$ 80:20 1 bar + 2 mM $^{15}NO_3^-$ + 20 mM $NO_3^-$ | 0,372 | 0,443 |

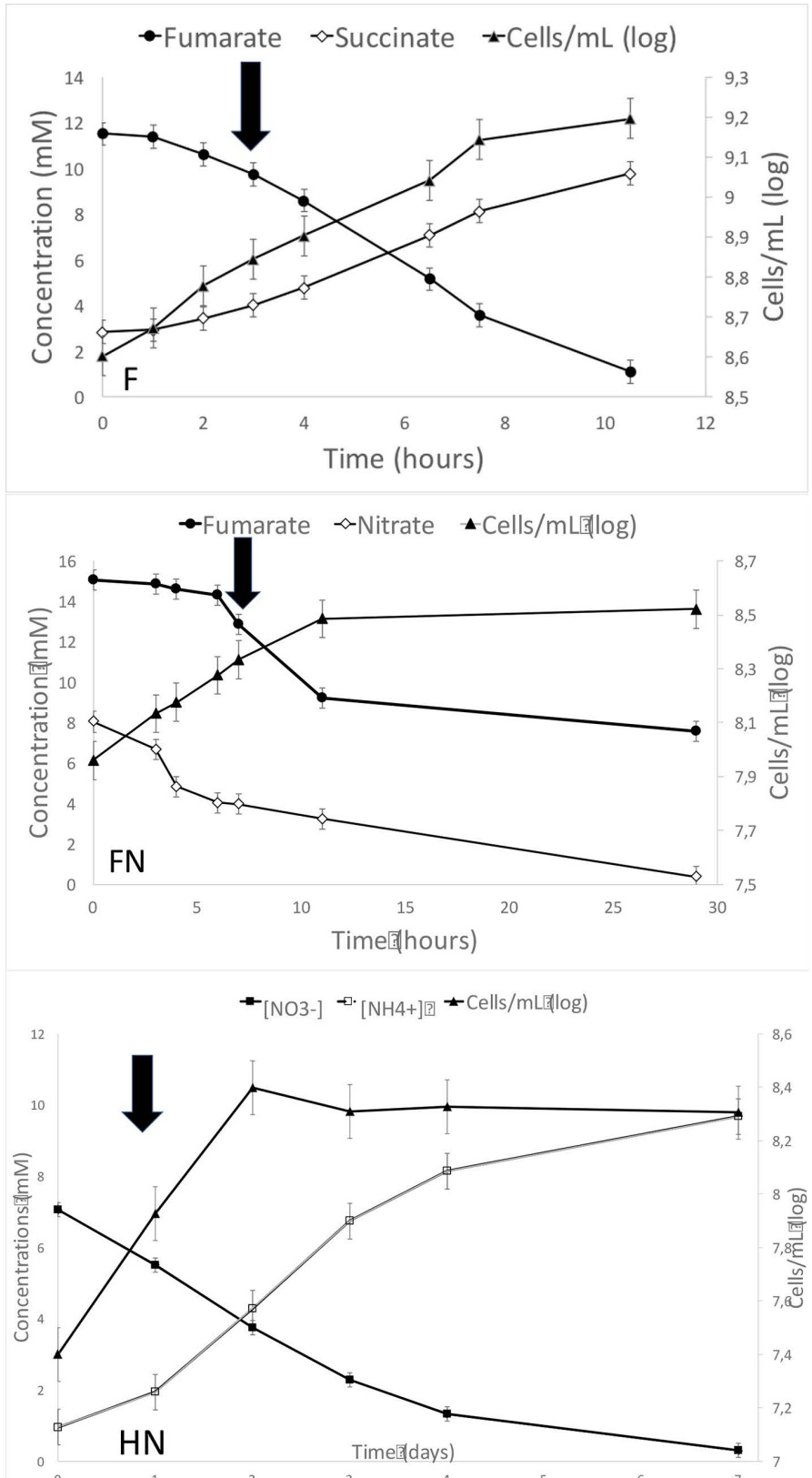

**Fig 6. Growth of strain V6Fe1[T] by fermentation or nitrate reduction.** Three conditions were tested: a) F: fermentation on fumarate alone used as the reference condition; b) FN: respiratory nitrate reduction with fumarate as

substrate; c) HN: nitrate reduction in autotrophic conditions with hydrogen as the sole electron donor. All these cultures were run in triplicate. In this figure, only one of these triplicates is shown for each condition. Error bars represent the standard deviation of the measurements. The black arrow indicates the time of sampling for RT-qPCR analysis.

intermediate) production was detected, as expected if denitrification occurred, while ammonium concentrations increased, indicative of the DNRA pathway. About one mole (0,89 to 0,99) of ammonium was produced per mole of nitrate consumed. Nitrate was also almost entirely consumed, as concentrations dropped from 7 mM to 0.3 mM. The DNRA reaction with dihydrogen can be written as $4 H_2 + NO_3^- + 2 H^+ \rightarrow NH_4^+ + 3 H_2O$. Nitrate reduction with dihydrogen as the sole electron donor was then exclusively performed by DNRA. In the exponential phase under our culture conditions (10 mM nitrate), only denitrification was observed with fumarate 15 mM as the carbon and energy source; only DNRA was observed with $H_2$/ $CO_2$ as the carbon and energy source. These results can be interpreted according to thermodynamic considerations. Free Gibbs energies of DNRA and denitrification can be computed (12) and show that DNRA is more favorable when hydrogen reacts with nitrate ($\Delta G° = $ -662 kJ/ mol $NO_3^-$ for DNRA versus $\Delta G° = $ -580 kJ/mol $NO_3^-$ for denitrification Free energy values for of denitrification and DNRA with fumarate as the electron donor are less conclusive ($\Delta G° = $ -694 kJ/mol $NO_3^-$ for DNRA versus $\Delta G° = $ -640 kJ/mol $NO_3^-$ for denitrification). DNRA yields more energy from one mole of nitrate, promoting this pathway when reducing power is in excess or limiting nitrate concentration.

On the contrary, denitrification should be favored in case of nitrate excess or limiting reducing power [72]. Therefore, the differential expression of the genes involved in each pathway was measured to test this hypothesis. The results of the quantitative RT-PCR are summarized in Fig 7.

Nitrate in the medium induced the expression of the nitrate reductase NapA. Gene expression was six times higher when nitrate was provided (FN and HN conditions) than in the F control condition. So, the high-affinity periplasmic nitrate reductase is the first step enzyme involved in nitrate reduction for both DNRA and denitrification. The expression of *cym*A seemed to be repressed when nitrate was added, especially in autotrophic conditions. In our experiments, transcription of *cym*A occurred only when fumarate was present in the medium. Thus, electrons should be transmitted to NapA *via* another electron shuttle. Indeed, the *cym*A expression pattern suggests it is instead involved in MK-7 oxidation coupled with fumarate reduction to succinate by the fumarate reductase (DSN97_05740, DSN97_05735) [73].

In FN conditions, when denitrification occurred, transcription of the *hcp* gene was 45 times higher than in the control condition F. Transcription of the *hcp* gene was relatively high in HN conditions, 4.4 times higher than in the control condition, when DNRA occurred. The Hcp role has long been debated [74]. Because its amino acid sequence resembles a carbon monoxide dehydrogenase that can reduce hydroxylamine, such a function has been proposed [74]. Hcp could also have a protective role against the toxicity of nitric oxide (NO), reaching high intracellular concentrations during nitrate reduction, especially denitrification [75,76].

The *nrf*A expression level was equivalent in F and FN conditions but twenty times higher in HN conditions. Thus, the pentaheme cytochrome c nitrite reductase (NrfA) may be the central enzyme catalyzing nitrite reduction to ammonium [77] in strain V6Fe1^T. Electrons for nitrite reduction in V6Fe1^T strain can be furnished by a hydrogenase encoded in the genome (DSN97_01270, DSN97_01760, DSN97_03610, DSN97_04825) via redox partners yet to be identified, perhaps a formate dehydrogenase encoded by DSN97_07560. NrfA-like cytochromes are also encoded in *D. essentukiensis* (LF845_RS03595), *D. desulfuricans*

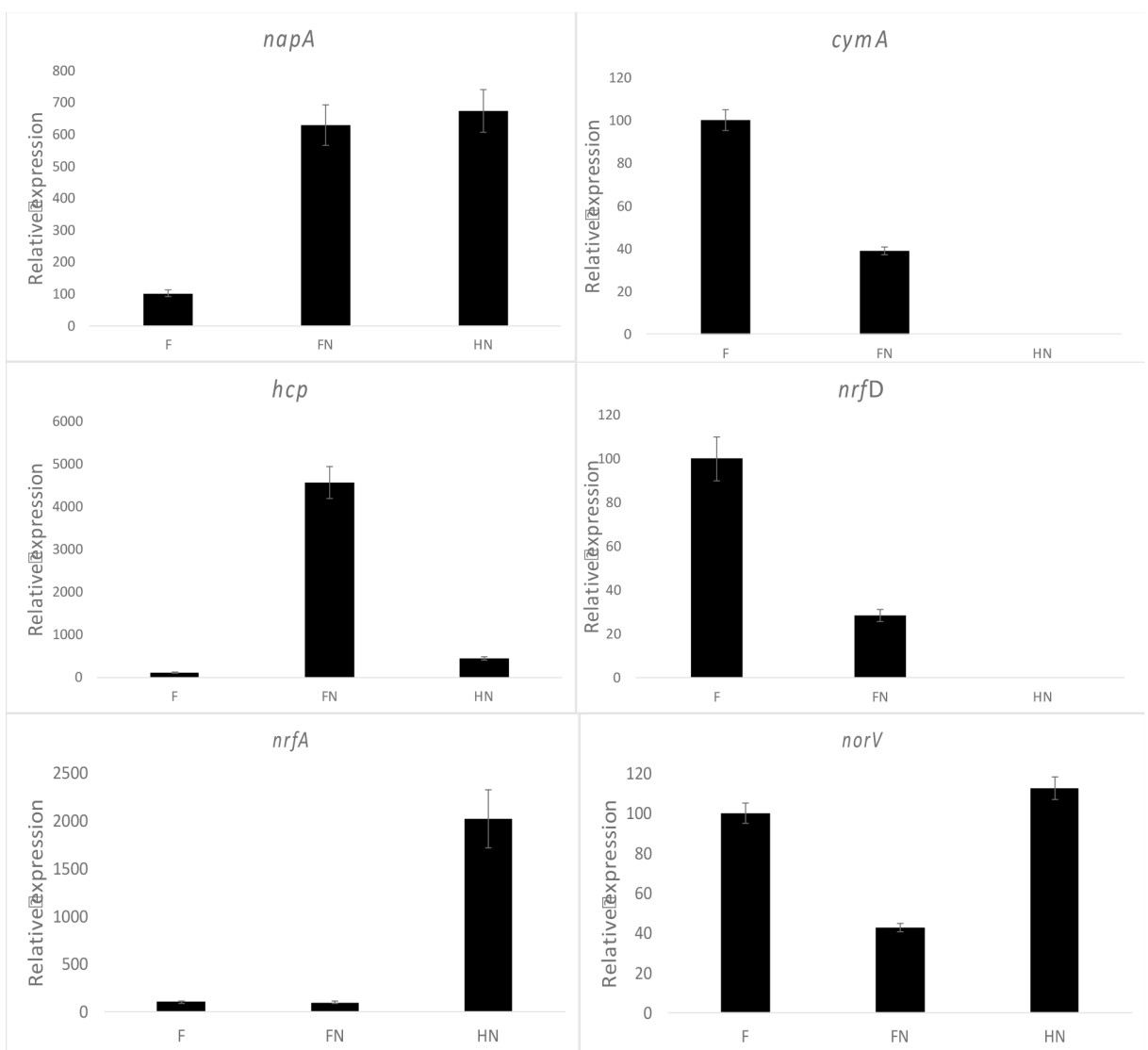

**Fig 7. Expression profiles of genes involved in nitrate reduction depending on the substrate.** Expression levels of all the genes were normalized to 16S rRNA gene expression. The RT-qPCR analyses were conducted on strain V6Fe1[T] at the exponential growth phase under different culture conditions. F represents the Fumarate control condition (100% expression), FN represents culture conditions with fumarate and nitrate, and HN represents the autotrophic culture condition with hydrogen as the sole electron donor and nitrate. The barplots represent differential expression of *nap*A (periplasmic nitrate reductase), *cym*A (tetraheme c-type cytochrome involved in anaerobic respiration), *hcp* (hybrid cluster protein, possibly involved in hydroxylamine reduction or nitric oxide detoxification), *nrf*D (transmembrane protein involved in energy conservation), *nrf*A (formate-dependent nitrite reductase), and *nor*V (automatically annotated as anaerobic nitric oxide reductase flavorubredoxin). Three cultures were performed in each condition (F, FN and HN), and three dilutions were used in each RT-qPCR assay.

(DEFDS_RS06970), and *D. autotrophicus* (FHQ18_RS01740) in a syntenic organization, but not in *C. nitroreducens* genome.

The *nor*V gene relative expression was equivalent in F and HN conditions and was down-regulated in FN conditions. It is annotated as a nitric oxide reductase and found in the genomes of *D. essentukiensis*, *D. autotrophicus*, *C. nitroreducens,* and *D. desulfuricans*. *E*ven these organisms do not use the denitrification pathway. Using the Blast algorithm, we found that this protein is classified as an FprA family A-type flavoprotein. Therefore, *nor*V

in strain V6Fe1$^T$ may encode a ferredoxin NADP+ reductase involved in other redox reactions. The relative expression of *nrf*D was maximal in the F condition. It was quasi-null in the HN conditions and intermediate in the FN condition. The periplasmic nitrite reductase (NrfABCD) was the first complex recognized to have a membrane subunit connected to the membrane through an integral membrane protein that interacts with quinones but does not have redox cofactors [78,79]. Sequence analyses indicated that NrfD homologs exist in many diverse enzyme families, such as polysulfide reductase (PsrABC) and other redox enzymes [79]. The reaction catalyzed by NrfD in strain V6Fe1$^T$ could thus be sulfur reduction, as these enzymes are annotated as polysulfide reductase and are repressed when nitrate is added to the medium. These results of the RT-qPCR experiments led to a conceptual model for nitrate reduction in strain V6Fe1$^T$, as represented in Fig 8. The switch between DNRA and denitrification in strain V6Fe1$^T$ is controlled by the balance between electron donor and electron acceptor availability, a large electron donor availability leading to the DNRA pathway [13].

## 5. Conclusion

Genomic and physiological analyses of strain V6Fe1$^T$ revealed several remarkable metabolic features of this bacterium within the *Deferribacterota* phylum. Genomic data suggested that $CO_2$ fixation occurs *via* an atypical, recently discovered, reversible oxidative TCA cycle [62, 63]. Nitrate was reduced by both DNRA and denitrification pathways, thus being reduced to ammonium or dinitrogen depending on the reducing power/nitrate concentration balance. The nitrate reduction system includes a soluble nitrate reductase NapA. The preference for NapA (over NarG) may be explained by the low nitrate concentrations in shallow hydrothermal systems. The conversion of nitrite to ammonium is mediated by the formate-dependent nitrite reductase NrfA. Adaptation is crucial in a rapidly changing environment such as the Vulcano hydrothermal system [80], and this can be achieved by efficiently regulating switches between versatile metabolic pathways as illustrated by the denitrification versus DNRA or the heterotrophic or autotrophic mode of the TCA cycle in strain V6Fe1$^T$. This strain is, therefore, an attractive model for studying the adaptation of bacteria to changing environments such as shallow hydrothermal vents.

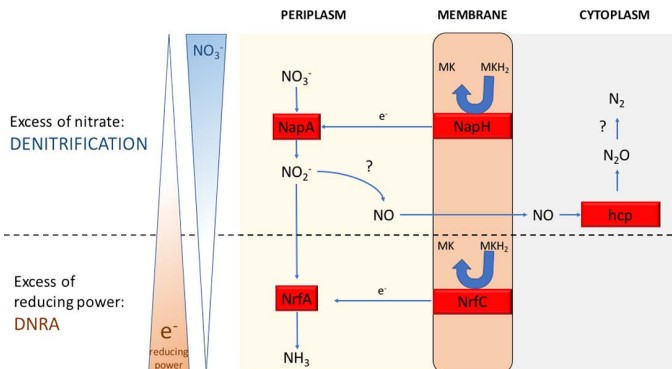

**Fig 8. The hypothetical enzymatic nitrate reduction system in strain V6Fe1$^T$ according to the nitrate/electron donor balance.** Strain V6Fe1$^T$ is unique within the *Deferribacterota* phylum by its ability to perform denitrification. NapA is the soluble periplasmic nitrate reductase characterized by its strong affinity for nitrate. We hypothesize that in case of excess of reducing power, strain V6Fe1$^T$ performs DNRA via a NrfA protein and that in case of shortage of reducing power or nitrate excess, strain V6Fe1$^T$ performs denitrification. Proteins involved in the denitrification process still deserve investigation.

## Proposition for a new species

Strain V6Fe1$^T$ and *Deferrivibrio essentukiensis* share many genomic traits and are closely related according to the 16S rRNA sequence (99.5% identity). Given that strain V6Fe1$^T$ can ferment proteinaceous substrates and organic acids, can grow lithoautotrophically with $H_2$/$CO_2$ as an energy and carbon source, harbors different fatty acids, and performs both denitrification and DNRA at the difference of its closest relatives, *Deferrivibrio essentukiensis* and *Petrothermobacter organivorans,* and based on their significant differences on growth optima (temperature, salinity, and pH), doubling time and genomic ANI value, we propose strain V6Fe1$^T$ ( = DSM 27501$^T$ = JCM 39088$^T$) to be the type strain of a novel species named *Deferrivibrio metallireducens*.

Description of *Deferrivibrio metallireducens* sp. nov. (me.tal'li.re.du'cens. L. n. metallum metal; L. part. adj. reducens leading back, bringing back, and, in chemistry, converting to a different oxidation state; N.L. part. adj. metallireducens reducing metal)

Small rods that are straight- to slightly vibrio-shaped. Cells are 4-8 μm long and 0.2-0.8 μm wide, with the Gram-negative type of cell wall. Non-motile. Anaerobic. Moderately thermophilic, growing between 45 and 65°C, with optimum growth at 55-60°C. Neutrophilic, growing between pH 5.1 and 7.7, with optimum growth at pH 6.3–6.5. Grows at NaCl concentrations ranging from 0 to 80 g/L, with optimum growth at 25 g/L NaCl. Capable of chemolithoautotrophic growth with hydrogen as an electron donor, manganese, insoluble ferric iron, elemental sulfur, or nitrate as electron acceptors, and $CO_2$ as a carbon source. Anaerobic oxidation of acetate, pyruvate, succinate, and proteinaceous substrates using sulfur, nitrate, Mn(IV), or insoluble Fe(III) as electron acceptors. Capable of fermentation yeast extract, peptone, and organic acids. Sensitive to ampicillin, bacitracin, chloramphenicol, cycloheximide, gentamicin, kanamycin, and tetracycline but resistant to vancomycin and streptomycin. Isolated on the seashore from sediments affected by hydrothermal circulations of the Vulcano hydrothermal system. The type strain is V6Fe1$^T$ ( = DSM 27501$^T$ = JCM 39088$^T$). The G + C content of its DNA is 34.8 mol %. Isolated from the Vulcano (38°24'45" N, 14° 57' 38" E) hydrothermal system.

## Supporting information

**Figure S1. Cladogram based on 16S rRNA gene sequences showing the position of *Deferrivibrio metallireducens* V6FE1T (ON783970) in the family *Deferrivibrionaceae*, related to the families *Deferribacteraceae* and *Geovibrionaceae* (Order *Deferribacterales*, Class *Deferribacteres*, Phylum *Deferribacterota*).**
The evolutionary history was inferred using the Maximum Likelihood method and General Time Reversible model. The percentage of trees in which the associated taxa clustered is shown next to the branches. This analysis involved 23 nucleotide sequences. The outgroup was composed of the following type species: *Thermanaeromonas toyohensis* ToBE (AB062280), *Moorella humiferrea* 64 FGQ (GQ872425), *Desulfovibrio indonesiensis* Ind 1 (Y09504), *Desulfovibrio marinus* E-2 (DQ365924), *Desulfonatronum cooperativum* Z-7999 (AY725424), *Desulfonatronum thioautotrophicum* ASO4-1 (FJ469577), *Desulfonatronum thiodismutans* MLF1 (AF373920), *Desulfonatronum lacustre* DSM 10312 (AF418171), *Desulfonatronum buryatense* Ki5 (KC417374). There were a total of 2753 positions in the final dataset. Evolutionary analyses were conducted in MEGA 7 (35). Taxonomy is based on LPSN.

**Figure S2. Polar lipids of strain V6Fe1T.**
The polar lipid profile of strain V6Fe1T consisted of glycolipids, phospholipids, phosphatidylglycerol, phosphatidylethanolamine, and diphosphatidylglycerol.

**Figure S3. Venn diagram of protein orthologs shared by the assembly of *D. autotrophicus* chromosome and plasmid of *D. desulfuricans* and *C. nitroreducens.***
Orthology analysis was performed with OrthoVenn2 web server using 0.01 blast e-value and 1.5 orthoMCL grain value. Numbers indicate shared or unique protein clusters and singleton proteins.

**Table S1. Primers used in this study, designed using Primer3 software version 4.1.0, targeting *cymA, hcp, napA, norV, nrfD* (present in two copies), *nrfA* genes encoding respectively a cytoplasmic membrane electron transport protein involved in nitrate reduction, a hydroxylamine reductase, a periplasmic nitrate reductase, a nitric oxide reductase, an integral transmembrane protein involved in the terminal transfer of electrons from the quinone pool into the terminal components of the Nrf pathway and a formate-dependent nitrite reductase.**

**Table S2. General features and genome sequencing information for *Deferrivibrio metallireducens* V6Fe1T according to MIGS recommendations.**

## Acknowledgments

We thank Isabelle Canihac for strain isolation, Amandine Ruiz and Manon Joseph-Bartoli for help in culture experiments, Sophie Guasco for assistance in qRT-PCR experiments, and Patricia Bonin and Patrick Raimbault for assistance in isotopic analyses. We thank Giorgio Capasso and Francesco Italiano of the Istituto Nazionale Geofisica e Vulcanologia (INGV) for their help in sampling and geochemical studies. The LABGeM (CEA/Genoscope & CNRS UMR8030), the France Génomique, and the French Bioinformatics Institute national infrastructures (funded as part of Investissement d'Avenir program managed by Agence Nationale pour la Recherche, contracts ANR-10-INBS-09 and ANR-11-INBS-0013) are acknowledged for support within the MicroScope annotation platform.

## Author contributions

**Conceptualization:** Grégoire Galès, Gaël Erauso.

**Data curation:** Grégoire Galès, Mélanie Hennart, Maverick Hannoun, Anne Postec, Gaël Erauso.

**Formal analysis:** Grégoire Galès, Anne Postec.

**Funding acquisition:** Gaël Erauso.

**Investigation:** Grégoire Galès, Maverick Hannoun.

**Methodology:** Grégoire Galès, Maverick Hannoun, Anne Postec, Gaël Erauso.

**Project administration:** Gaël Erauso.

**Resources:** Gaël Erauso.

**Software:** Grégoire Galès, Mélanie Hennart, Maverick Hannoun, Anne Postec.

**Validation:** Grégoire Galès, Maverick Hannoun.

**Visualization:** Mélanie Hennart.

**Writing – original draft:** Grégoire Galès, Anne Postec, Gaël Erauso.

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
