## [Decision Letter · Decision Letter 0]

13 Sep 2024

PONE-D-24-31234Metabolic versatility and nitrate reduction pathways of a new thermophilic bacterium of the Deferrivibrionaceae: Deferrivibrio metallireducens sp. nov isolated from hot sediments of Vulcano Island, ItalyPLOS ONE

Dear Dr. GALES,

Thank you for submitting your manuscript to PLOS ONE. After careful consideration, we feel that it has merit but does not fully meet PLOS ONE’s publication criteria as it currently stands. Therefore, we invite you to submit a revised version of the manuscript that addresses the points raised during the review process.When you're ready to submit your revision Oct 28 2024 11:59PM, log on to https://www.editorialmanager.com/pone/ and select the 'Submissions Needing Revision' folder to locate your manuscript file.

We look forward to receiving your revised manuscript.

Kind regards,

Vasu D. Appanna

Academic Editor

PLOS ONE

Additional Editor Comments:

Dear Dr. Gales

Your manuscript titled "Metabolic versatility and nitrate reduction pathways of a new thermophilic bacterium of the Deferrivibrionaceae: Deferrivibrio metallireducens sp. nov isolated from hot sediments of Vulcano Island, Italy" has been reviewed and has been found acceptable for publication with minor revision. The work has revealed some interesting metabolic attributes of this thermophilic bacterium, however, it is important to limit the discussion section to the results obtained. Please take into account the suggestions of the reviewers in regard to the points raised while framing the revised version of the manuscript.

Thank you

Vasu D. Appanna PhD

Professor

Reviewers' comments:

Reviewer's Responses to Questions

**Comments to the Author**

1. Is the manuscript technically sound, and do the data support the conclusions?

Reviewer #1: Yes

Reviewer #2: Yes

2. Has the statistical analysis been performed appropriately and rigorously? 

Reviewer #1: N/A

Reviewer #2: N/A

3. Have the authors made all data underlying the findings in their manuscript fully available?

Reviewer #1: Yes

Reviewer #2: Yes

4. Is the manuscript presented in an intelligible fashion and written in standard English?

Reviewer #1: Yes

Reviewer #2: Yes

5. Review Comments to the Author

Reviewer #1: The article by Grégoire et al. describes the isolation of a novel thermophilic bacterium.

The manuscript is of interest, and the data support the conclusions.

I only have minor recommendations:

- Figure 6, FN graph: It is difficult to discern between the groups. Please change one to an open box or something more recognizable.

- Figure 7: The X axis is cut off for cymA

- Figure captions: What is your n value for the studies? Please list in figure captions (for all figures). For the TEM, was this a representative image? How many images were captured?

- The Results and Discussion section is quite long (90 references) and contains a lot of hand-waving. For example, you discuss the roTCA cycle, and other enzymes that "could be" at work, but provide no functional evidence for many of them. I recommend trimming this section down by 2-3 pages and focusing on what was discovered and not what you believe may be happening

Reviewer #2: Dear Editor,

I have thoroughly reviewed the manuscript entitled "Metabolic versatility and nitrate reduction pathways of a new thermophilic bacterium of the Deferrivibrionaceae: Deferrivibrio metallireducens sp. nov. isolated from hot sediments of Vulcano Island, Italy," and I would like to offer the following comments for improvement: I have also mentined the some changes in the MS pdf.

*Please remove any unnecessary paragraphs from the abstract for better clarity.

*Ensure consistent formatting for temperature symbols (Celsius) throughout the manuscript.

*Properly align the tables and address any missing entries.

*Avoid the use of capitalized words inappropriately throughout the manuscript. Additionally, the manuscript requires English language corrections.

*Ensure proper referencing style and maintain consistency in the reference format.

*Italicize all scientific names and follow the standard rules of scientific nomenclature.

*In the Materials & Methods section, include the concentrations of the PCR components and provide details of the PCR programs used.

*Consider using iTOL tools for better representation of the phylogenetic analysis figure.

*Correct any grammatical and language errors throughout the manuscript.

6. PLOS authors have the option to publish the peer review history of their article (what does this mean? ). If published, this will include your full peer review and any attached files.

**Do you want your identity to be public for this peer review?** For information about this choice, including consent withdrawal, please see our Privacy Policy .

Reviewer #1: No

Reviewer #2: No

---

## [Author Response · Author response to Decision Letter 0]

28 Oct 2024

Dear Professor Appanna,

I am grateful to you and the two reviewers for your contributions. As a result, we have improved the quality and relevance of our article. The publisher and the two reviewers were correct in noting that the results and discussion section was too long and speculative. We have reduced the article by two or three pages and withdrawn ten references. We have focused on what we can demonstrate and eliminated hand-waving. We are confident you will find the corrections to the manuscript convincing.

You will find the answers to your questions and comments here, as well as in the text and figures. We have made the document easier to read by using bold and underlined text for our answers.

We have taken great care to ensure that the manuscript will meet your journal's expectations.

This section is irrelevant to our manuscript. We simply benefited from the help of Italian colleagues, who are mentioned in the acknowledgments.

No permits were required, as Italy is not a signatory to the Nagoya Protocol. We introduced the M&M section by saying: “Although Italy is not a signatory to the Nagoya Protocol, we carried out the sampling with the authorization of our collaborators from the INGV Palermo (Italy), as stated in the acknowledgments.”

Indeed, we (Galès, Postec & Erauso) published last year a strain characterization in IJSEM from comparable samples:” Marinitoga aeolica sp. nov., a novel thermophilic anaerobic heterotroph isolated from a shallow hydrothermal field of Panarea Island in the Aeolian archipelago, Italy.”

It has been done.

It has been done. No cited article has been retracted yet. As requested by the first reviewer, we have reduced the list of references (ten references withdrawn).

Reviewer #1: The article by Grégoire et al. describes the isolation of a novel thermophilic bacterium.

The manuscript is of interest, and the data support the conclusions.

I only have minor recommendations:

- Figure 6, FN graph: It is difficult to discern between the groups. Please change one to an open box or something more recognizable.

It has been done. Nitrate is shown in an open box. This makes the figure easier to understand.

- Figure 7: The X axis is cut off for cymA

It has been corrected.

- Figure captions: What is your n value for the studies? Please list in figure captions (for all figures). For the TEM, was this a representative image? How many images were captured?

All experiments were performed in triplicate. About 25 TEM images were taken, from which we selected Figure 3. For the RT-qPCR experiments, three dilutions were analyzed. These clarifications have been included in the text and figure legends.

- The Results and Discussion section is quite long (90 references) and contains a lot of hand-waving. For example, you discuss the roTCA cycle, and other enzymes that “could be” at work, but provide no functional evidence for many of them. I recommend trimming this section down by 2-3 pages and focusing on what was discovered and not what you believe may be happening

The Results and Discussion section has been significantly reduced, as the reviewer asked. We have removed all sections deemed too speculative. The relevant section is now more than two pages shorter than initially planned, which improves the manuscript. Ten references have been withdrawn. We have eliminated the lengthy discussion of the roTCA cycle and the denitrification/DNRA switch regulation, which will be the subject of further research.

Reviewer #2: Dear Editor,

I have thoroughly reviewed the manuscript entitled "Metabolic versatility and nitrate reduction pathways of a new thermophilic bacterium of the Deferrivibrionaceae: Deferrivibrio metallireducens sp. nov. isolated from hot sediments of Vulcano Island, Italy," and I would like to offer the following comments for improvement: I have also mentined the some changes in the MS pdf.

*Please remove any unnecessary paragraphs from the abstract for better clarity.

There have been none in accordance with the wishes of the first reviewer.

*Ensure consistent formatting for temperature symbols (Celsius) throughout the manuscript.

It has been done throughout the text.

*Properly align the tables and address any missing entries.

This has been done.

*Avoid the use of capitalized words inappropriately throughout the manuscript. Additionally, the manuscript requires English language corrections.

This has been corrected.

*Ensure proper referencing style and maintain consistency in the reference format.

This has been done.

*Italicize all scientific names and follow the standard rules of scientific nomenclature.

This has been done.

*In the Materials & Methods section, include the concentrations of the PCR components and provide details of the PCR programs used.

This has been done.

In the text: “Quantitative PCR was performed with 2 L cDNA (directly reverse transcribed DNA and 10-1/10-2 dilutions), 0.25 L of each 10 M primer, 10 L of 2X SyberGreen and 7.5 L of ultrapure RNAse/DNAse-Free distilled water (Thermofisher). Thermal cycling conditions were as follows: 2 min at 98°C, followed by 30 cycles of 5 s at 98°C, 30 s at 61°C, and 20 s at 72°C. Data collection was performed during each annealing phase.”

*Consider using iTOL tools for better representation of the phylogenetic analysis figure.

Thank you for your valuable feedback regarding the phylogenetic analysis figures. We want to clarify that the phylogenetic trees presented in Figures 1 and 2 have already been generated using iTOL. To enhance clarity and provide a more detailed response to the reviewer, we have updated the legends for both figures accordingly.

*Correct any grammatical and language errors throughout the manuscript.

This has been done, we hope our efforts will be seen and appreciated.

We hope you will enjoy reading the corrections to this article. If you have any questions, please don’t hesitate to contact us.

The corresponding authors

Grégoire Galès

gregoire.gales@univ-amu.fr

Gaël Erauso

gael.erauso@univ-amu.fr

---

## [Decision Letter · Decision Letter 1]

21 Nov 2024

Metabolic versatility and nitrate reduction pathways of a new thermophilic bacterium of the Deferrivibrionaceae: Deferrivibrio metallireducens sp. nov isolated from hot sediments of Vulcano Island, Italy

PONE-D-24-31234R1

Dear Dr. Gales

We’re pleased to inform you that your manuscript has been judged scientifically suitable for publication and will be formally accepted for publication once it meets all outstanding technical requirements.

Kind regards,

Vasu D. Appanna

Academic Editor

PLOS ONE

Additional Editor Comments (optional):

Reviewers' comments:

Reviewer's Responses to Questions

**Comments to the Author**

1. If the authors have adequately addressed your comments raised in a previous round of review and you feel that this manuscript is now acceptable for publication, you may indicate that here to bypass the “Comments to the Author” section, enter your conflict of interest statement in the “Confidential to Editor” section, and submit your "Accept" recommendation.

Reviewer #1: All comments have been addressed

2. Is the manuscript technically sound, and do the data support the conclusions?

Reviewer #1: (No Response)

3. Has the statistical analysis been performed appropriately and rigorously? 

Reviewer #1: (No Response)

4. Have the authors made all data underlying the findings in their manuscript fully available?

Reviewer #1: (No Response)

5. Is the manuscript presented in an intelligible fashion and written in standard English?

Reviewer #1: (No Response)

6. Review Comments to the Author

Reviewer #1: (No Response)

7. PLOS authors have the option to publish the peer review history of their article (what does this mean? ). If published, this will include your full peer review and any attached files.

**Do you want your identity to be public for this peer review?** For information about this choice, including consent withdrawal, please see our Privacy Policy .

Reviewer #1: No

---

## [Editor Report · Acceptance letter]

PONE-D-24-31234R1

PLOS ONE

Dear Dr. Grégoire,

I'm pleased to inform you that your manuscript has been deemed suitable for publication in PLOS ONE. Congratulations! Your manuscript is now being handed over to our production team.

Kind regards,

on behalf of

Dr. Vasu D. Appanna

Academic Editor

PLOS ONE